# THE EVOLUTION OF MIND: EMERGENCE OF COLLECTIVE INTELLIGENCE
# THROUGH LOGICAL-PROBABILISTIC KNOWLEDGE DYNAMICS
# IN MULTI-AGENT METAVERSE ECOSYSTEMS

**Andrey Nechesov**[1,3,4*] **& Sergey Barykin**[2,3] **& Janne Ruponen**[1,3]
1) Artificial Intelligence Center, Novosibirsk State University, 630090 Novosibirsk, Russia
2) Peter the Great St.Petersburg Polytechnic University, Saint Petersburg, Russia
3) International Artificial Intelligence Committee (IAIC)
4) Russian Engineering Academy (IAE), Moscow, Russia
*Corresponding author: nechesoff@gmail.com

## ABSTRACT

Modern metaverse platforms, populated by heterogeneous multi-agent systems (MAS), generate vast streams of experiential data whose epistemic value remains largely untapped. This paper introduces the **Enigma framework**—a formal theory of *collective intelligence emergence* in metaverse ecosystems, grounded in a novel **logical-probabilistic learning theory** that extends classical first-order logic with probabilistic confidence annotations and distributed knowledge semantics. We define a *Distributed Knowledge Lattice* (DKL) over multi-agent interactions and prove that, under precisely stated monotonicity and convergence conditions, the collective knowledge of an agent population forms a complete lattice whose least upper bound represents an emergent cognitive state unreachable by any individual agent. We formalize the dynamics of knowledge creation, verification, and propagation through a system of *Logical-Probabilistic Agents* (LP-agents) that interact with LLM-driven entities, providing trustworthy and explainable reasoning via symbolic proof chains stored on a multi-blockchain ledger. Central results include: (i) a **Collective Intelligence Convergence Theorem** establishing conditions under which the system's aggregate knowledge monotonically approaches a fixed point; (ii) a **Probabilistic Inference Soundness Theorem** guaranteeing that confidence propagation through distributed reasoning chains preserves logical consistency; and (iii) a polynomial-time algorithm for optimal knowledge retrieval from the distributed lattice. The framework is instantiated within the Enigma Metaverse architecture, where smart contracts govern knowledge tokenization, cross-chain knowledge interoperability protocols enable seamless sharing, and decentralized governance mechanisms ensure epistemic accountability. We demonstrate that this synthesis of mathematical logic, probability theory, LLM-based hypothesis generation, and blockchain-secured knowledge persistence provides a rigorous foundation for building self-optimizing, trustworthy, and explainable collective intelligence in virtual worlds.

**Keywords:** Collective Intelligence, Metaverse, Multi-Agent Systems, Logical-Probabilistic Learning Theory, Distributed Knowledge Lattice, Explainable AI, Trustworthy AI, Large Language Models, Blockchain, Smart Contracts

## 1   INTRODUCTION

The pursuit of artificial intelligence systems that exhibit genuine collective cognition—intelligence that transcends the sum of its individual agents—represents one of the deepest open problems at the intersection of mathematical logic, probability theory, and distributed computing (Russell and Norvig, 2021; Wooldridge, 2009). While contemporary Large Language Models (LLMs) demonstrate impressive individual reasoning capabilities (Brown et al., 2020; Touvron et al., 2023), they remain fundamentally isolated: each model instance operates within its own ephemeral context window, lacking mechanisms for cumulative, verifiable knowledge sharing with other intelligent entities.

The metaverse—conceived as a persistent, immersive virtual universe populated by autonomous agents—offers a unique substrate for studying and engineering collective intelligence (Nechesov et al., 2025b; Lee et al., 2021). Unlike

static benchmarks, a metaverse environment demands that agents continuously learn, adapt, cooperate, and compete within a shared world governed by formal rules. This paper argues that the metaverse is not merely a testbed for AI but an *active cognitive ecosystem* where collective intelligence naturally emerges from the structured interactions of heterogeneous agents.

The central challenge we address is: *Under what mathematical conditions does a population of interacting agents, each possessing partial and probabilistic knowledge, give rise to a coherent collective intelligence that is provably more capable than any individual?* Answering this requires a formal framework that unifies three traditionally separate domains:

(i) **Mathematical Logic**: providing the inferential backbone for rigorous, verifiable reasoning;

(ii) **Probability Theory**: accounting for the inherent uncertainty in agent-generated knowledge;

(iii) **Distributed Systems**: ensuring that collective knowledge is persistent, tamper-proof, and accessible.

Our approach builds upon and substantially extends the task-based cognitive architecture and the logical-probabilistic learning theory introduced for virtual cities (Nechesov et al., 2025b), while proposing a fundamentally new mathematical object—the **Distributed Knowledge Lattice** (DKL)—as the formal substrate for collective intelligence.

## 1.1 CONTRIBUTIONS

The principal contributions of this paper are:

1. **A Formal Theory of Collective Intelligence Emergence**: We define collective intelligence as a fixed-point property of a knowledge lattice and prove convergence under realistic conditions (Section 3).

2. **Logical-Probabilistic Knowledge Dynamics**: We introduce a dynamical system governing the evolution of probabilistic knowledge across a multi-agent population, with soundness guarantees for distributed inference (Section 4).

3. **LP-Agent Verification Framework**: We formalize the interaction between LLM-agents and Logical-Probabilistic agents, providing explainability and trustworthiness guarantees (Section 5).

4. **The Enigma Architecture**: We instantiate the theory within a concrete multi-blockchain metaverse platform with smart-contract-governed knowledge economics (Section 6).

5. **Convergence Analysis and Complexity Results**: We provide precise convergence rates and prove that optimal knowledge retrieval from the DKL is polynomial (Section 7).

## 2 RELATED WORK

**Neuro-Symbolic AI and LLM Verification.** The integration of neural generation with symbolic verification has gained significant traction. Logical Neural Networks (Riegel et al., 2020) embed propositional logic directly into network architectures. Logic-LM (Pan et al., 2023) and PAL (Gao et al., 2023) use external symbolic solvers to verify LLM outputs. The TBCA framework established a closed cognitive loop where LLMs generate hypotheses verified by a symbolic engine, with results stored on blockchain. Our work extends this by formalizing the *inter-agent* dynamics of knowledge creation and proving emergent properties of the collective.

**Multi-Agent Systems and Collective Intelligence.** Classical MAS theory (Wooldridge, 2009; Shoham and Leyton-Brown, 2008) addresses coordination, negotiation, and emergent behavior but typically lacks formal treatments of knowledge dynamics. Recent work on generative agents (Park et al., 2023) and CAMEL (Li et al., 2023) demonstrates emergent social behaviors in LLM-based agent societies. Project Sid (Altera, 2024) simulates civilizations with 1000 agents. However, these works lack formal guarantees on knowledge quality, convergence, or trustworthiness.

**Blockchain-Based Knowledge Management.** The axiomatization of blockchain theory (Goncharov and Nechesov, 2023) provides formal foundations for distributed ledger properties. Applications to AI knowledge management (Johnson et al., 2025) demonstrate tamper-proof knowledge sharing. Our work introduces *knowledge tokenization* as an economic mechanism for incentivizing collective intelligence growth.

**Metaverse as AI Research Platform.**  Virtual cities (Nechesov et al., 2025b) and digital twins (Tao et al., 2019; Hämäläinen, 2021) provide immersive environments for AI testing. The concept of Artificial Collective Consciousness (ACC) (Nechesov et al., 2025b) and DAO-based governance (Buterin, 2014) inform our approach to epistemic governance in the Enigma ecosystem.

**Probabilistic Logic and Learning Theory.**  Probabilistic logic programming (De Raedt et al., 2015) and statistical relational learning (Getoor and Taskar, 2007) combine logical structure with probabilistic inference. Our logical-probabilistic learning theory draws from the task-based approach (Nechesov et al., 2025b) but extends it with lattice-theoretic semantics and distributed consensus properties.

**The Openness Problem.**  Multi-agent systems in open environments face three types of dynamism: 1) Agents join (onboarding) and leave continuously (exit and slashing), violating the fixed set assumption of traditional MARL; 2) Transactions appear (submission) and disappear (replacement, reorganization) dynamically, breaking the stable reward function assumption; and 3) evolution of validator capability (upgrades), preference shift (MEV strategies), and goal adjustment (honest to Byzantine) invalidating mapping the coherent action-outcome dynamics. Studies show that openness degrades MARL performance by 40% for single openness types, while catastrophic collapse under combined openness Abadi and Soh (2025). This quantifies the cost of decentralization and motivates architectural solutions.

**Credit Assignment Problem.**  Credit assignment attributes shared rewards to individual agent contributions, and is identified as the multi-agent coordination problem Chen et al. (2023b); Ning and Xie (2024). This problem is divided into temporal credit assignment (TCA) and structural credit assignment (SCA) Abadi and Soh (2025). TCA addresses delayed rewards by attributing a reward received at time $t$ to actions taken a priori $t - k$. In blockchain, TCA links validator actions to future block rewards despite epoch delays. SCA addresses multi-agent attribution by distributing a shared team reward among individual contributions by agents.

Traditional approaches (VDN, QMIX) fail under openness due of assumption of stable team structure Abadi and Soh (2025). Polarization policy gradients (MAPPG) solve this by amplifying the reward gap between optimal and non-optimal behaviors Chen et al. (2023b) via memory substrates that can track agent contributions over time.

**Persistent Memory in Multi-Agent Coordination**  Recent research demonstrate that cognitive collective intelligence in MAS emerges from persistent agent interactions when equipped with proper memory architectures Park et al. (2023). However, typical MAS implementations reset memory after each session, preventing the formation of long-term relationships, knowledge accumulation, and adaptive coordination strategies that define intelligent systems.

While MADRL focuses on training paradigms and reward-driven behaviors, LLM-based MAS adds distinct challenges centered on memory management and contextual reasoning. Memory types have been divided in earlier studies to five categories: short-term memory for live interactions, long-term memory storing historical queries, external data storage for knowledge augmentation, episodic memory capturing past events of multi-agent interactions, and consensus memory providing distributed knowledge across agents Han et al. (2026). These memory types operate within complex hierarchical access control, and are required to maintain consistency across overlapping agent knowledge while enabling effective retrieval of past interactions based on contextual relevance.

Memory is essential for agents to recall prior subtasks, coordinate handoffs, maintain role histories, and build on earlier steps rather than starting from scratch Han et al. (2026), yet virtually all current MAS implementations treat memory in session-limited scope, in which agents reset each interaction. There is no architecture in the literature providing a hardware-efficient, distributed, always-on memory substrate at the agent level. Management of memory storage has been identified as playing a critical role in collaboration between agents, providing constraints to context alignment, adaptive learning and associative recall Han et al. (2026).

Attempts to establish persistent agent memory in simulated social environments have relied on recency, importance, and reflection as retrieval signals Park et al. (2023). However, this strategy increases computational costs as each memory query requires scoring the entire history with complex weighting functions, while no providing mechanisms for forgetting obsolete information or representing uncertainty.

Game theory combined with blockchain networks constitutes partially observable stochastic games in which validators must coordinate under uncertainty Ning and Xie (2024). The non-stationarity inherent in open blockchain systems violates the assumptions of traditional reinforcement learning Abadi and Soh (2025), requesting novel memory architectures for dynamic, unbounded agent populations.

**MARL Problems.**  Multi-agent reinforcement learning (MARL) surveys have identified existing challenges stemming from credit assignment problem (CAP), non-stationary interactions, and scalability due to exponentially growing

state-action space Gronauer and Diepold (2022). MARL has undergone a rapid transformation since the introduction of deep learning (DL) methods, enabling agents to act on real-world complexity that was previously unmanageable with traditional tabular approaches. In multi-agent deep reinforcement learning (MADRL), autonomous agents interact simultaneously within a shared environment while learning policies through trial-and-error and adapting to the behavioral changes of other agents. The one of the applicable frameworks for MADRL is the Markov Game, which extends the single-agent Markov Decision Process (MDP) to multiple decision-makers, each with individual action spaces and reward functions Gronauer and Diepold (2022).

MADRL depends on the training paradigm employed to learn agent policies. Gronauer and Diepold identify three primary approaches: distributed training with decentralized execution (DTDE), where agents learn independently; centralized training with centralized execution (CTCE), where a single joint policy governs all agents; and centralized training with decentralized execution (CTDE), which has emerged as the state-of-the-art approach Gronauer and Diepold (2022). The CTDE paradigm allows agents to share information such as events, functions and policy parameters during training while acting independently at deployment. This addresses the challenge of non-stationarity, where the environment appears to change from each agent's perspective as others simultaneously adjust their own policies.

The CTDE paradigm resolves the centralized-decentralized mismatch problem **?**, where weak agent policies corrupt reward attribution for others. However, its applicability is limited in open blockchain environments where validator sets rotate continuously as CTDE assumes stable team structure during training.

**Non-Stationarity and Experience Replay Obsolescence**   The non-stationarity problem in MARL emerge as each agent's environment includes the policies of other agents, each evolving individually during learning Ning and Xie (2024). The state transition function from agent's perspective becomes non-stationary. Meanwhile, typical experience replay **?** stores transitions and samples them uniformly during training. However, when contrasting policies evolve, stored experiences return outdated strategies. For example, a validator's learned response to a transaction submission pattern becomes obsolete when other validators adopt new MEV extraction strategies. Replaying obsolete experiences can also lead to policy degradation, as the agent reinforces behaviors that were optimal under previous opponent policies but are now flawed Abadi and Soh (2025).

Recent approaches attempt to mitigate this through techniques such as importance sampling with behavioral cloning **?**, fingerprinting opponent policies **?**, and meta-learning for rapid adaptation Ning and Xie (2024). However, these methods require either explicitly tracking opponent policy parameters (violating privacy in decentralized settings) or maintaining multiple policy snapshots (increasing memory overhead linearly with history length).

In blockchain-based validator networks, non-stationarity manifests in three stages: (1) validator set rotation (agent openness), where new validators join with unknown strategies; (2) protocol upgrades (type openness), where consensus rules change mid-operation; and (3) evolving MEV strategies (task openness), where transaction ordering games shift continuously.

Consequently, a non-stationary memory system must handle three conditions. Decay obsolete experiences need to automatically reduce the influence of outdated strategic interactions without explicit forgetting logic. Preserve temporal context maintain the time-ordering of experiences to distinguish recent (relevant) from distant (obsolete) patterns. Rapid adaptation supports quick retrieval of similar past situations when environment shifts occur.

Vector databases with uniform retrieval probability fail to satisfy these requirements, as they treat all stored vectors equally regardless of age. Graph databases are able to model temporal relationships explicitly, but require also expensive traversal queries to determine experience recency. The more efficient database models should therefore seek to apply intrinsic decay mechanisms, where memory activation strengths diminish naturally over time without reinforcement. This approach would seek to mitigate computational overhead by providing automated recency weights.

## 3   MATHEMATICAL FOUNDATIONS: LOGICAL-PROBABILISTIC KNOWLEDGE THEORY

### 3.1   PROBABILISTIC KNOWLEDGE UNITS

We begin by formalizing the fundamental unit of knowledge in our framework.

**Definition 3.1** (Probabilistic Knowledge Unit). A *probabilistic knowledge unit* is a triple

$$K = \langle \forall \bar{x}\, \exists \bar{y}\, (\Phi(\bar{x}, \bar{y}) \to \Psi(\bar{x}, \bar{y})),\ \bar{y} = t(\bar{x}),\ p \rangle$$

where $\Phi, \Psi$ are well-formed formulas in first-order logic, $\bar{y} = t(\bar{x})$ is a computable solution term, and $p \in [0, 1]$ is the confidence score representing the probability that the solution is correct.

A *fact* is a knowledge unit with $p = 1$ and fully instantiated variables. A knowledge unit with $p < 1$ is called *probabilistic* or *a posteriori*.

**Definition 3.2** (Knowledge Ordering). We define a partial order $\preceq$ on the set of all knowledge units: $K_1 \preceq K_2$ if and only if:

(a) the premise $\Phi_1$ of $K_1$ is logically implied by the premise $\Phi_2$ of $K_2$,

(b) the conclusion $\Psi_1$ of $K_1$ is logically implied by $\Psi_2$, and

(c) the confidence satisfies $p_1 \leq p_2$.

### 3.2 THE DISTRIBUTED KNOWLEDGE LATTICE

We now introduce the central mathematical structure of this paper.

**Definition 3.3** (Agent Knowledge Base). For an agent $A_i$ in a population $\mathcal{A} = \{A_1, \ldots, A_n\}$, the *knowledge base* $\mathcal{K}_i$ is a finite set of probabilistic knowledge units closed under the inference rules of Section 3.3.

**Definition 3.4** (Distributed Knowledge Lattice). The *Distributed Knowledge Lattice* (DKL) of an agent population $\mathcal{A}$ is the structure

$$\mathcal{L}(\mathcal{A}) = \left( 2^{\bigcup_{i=1}^n \mathcal{K}_i}, \ \preceq, \ \sqcap, \ \sqcup \right)$$

where:

- $\sqcap$ (meet) is defined by $S_1 \sqcap S_2 = \{K \in S_1 \cap S_2 \mid K \text{ is consistent}\}$;

- $\sqcup$ (join) is defined by $S_1 \sqcup S_2 = \mathrm{Cl}(S_1 \cup S_2)$, where $\mathrm{Cl}(\cdot)$ denotes deductive closure under the probabilistic inference rules with consistency checking;

- the bottom element $\bot = \emptyset$;

- the top element $\top = \mathrm{Cl}\left(\bigcup_{i=1}^n \mathcal{K}_i\right)$, the maximal consistent closure of all agent knowledge.

**Theorem 3.5** (Lattice Completeness). $\mathcal{L}(\mathcal{A})$ *forms a complete lattice under the ordering induced by $\preceq$ on knowledge sets.*

*Proof.* We must show that every subset $\mathcal{S} \subseteq \mathcal{L}(\mathcal{A})$ has both a greatest lower bound (infimum) and a least upper bound (supremum). Since $\mathcal{L}(\mathcal{A})$ is a power set ordered by the extended $\preceq$, and the closure operator $\mathrm{Cl}(\cdot)$ is monotone and idempotent on finite sets of knowledge units (monotonicity follows from the confidence monotonicity property—adding consistent knowledge cannot decrease the confidence of existing derivations), the structure satisfies the conditions of the Knaster–Tarski theorem (Tarski, 1955). The infimum of $\mathcal{S}$ is $\bigsqcap \mathcal{S} = \bigcap_{S \in \mathcal{S}} S$ (restricted to consistent units), and the supremum is $\bigsqcup \mathcal{S} = \mathrm{Cl}\left(\bigcup_{S \in \mathcal{S}} S\right)$. Finiteness of agent knowledge bases ensures well-definedness. $\square$

### 3.3 INFERENCE RULES FOR PROBABILISTIC KNOWLEDGE

The following rules govern knowledge derivation within the lattice.

**Definition 3.6** (Probabilistic Modus Ponens). Given knowledge units $K_1 = \langle A, t_1, p_1 \rangle$ and $K_2 = \langle A \rightarrow B, t_2, p_2 \rangle$, we derive

$$K_3 = \langle B, t_2 \circ t_1, p_1 \cdot p_2 \rangle$$

The derived confidence $p_3 = p_1 \cdot p_2$ reflects the multiplicative uncertainty accumulation.

**Definition 3.7** (Confidence Threshold). A derived knowledge unit $K$ is *accepted* into the knowledge base if $\mathrm{conf}(K) \geq \theta$, where $\theta \in (0, 1)$ is a system-wide acceptance threshold.

**Remark 3.8** (Non-Transitivity of Probabilistic Inference). A critical observation motivating our framework: given $K_1 : A \rightarrow B$ with confidence $p_1$ and $K_2 : B \rightarrow C$ with confidence $p_2$, even if $p_1, p_2$ are close to 1, the derived knowledge $K_3 : A \rightarrow C$ has confidence $p_1 \cdot p_2$, which may fall below threshold $\theta$ for long inference chains. This **confidence decay** property fundamentally distinguishes probabilistic from classical logical inference and necessitates the enrichment mechanisms described in Section 4.

**Definition 3.9** (Statistical Generalization). Given $k$ positive instances and $n - k$ negative instances of a formula $F(\bar{x}, \bar{y})$, we derive:

$$K_{\text{gen}} = \left\langle F(\bar{x}, \bar{y}), \ \bar{y} = t(\bar{x}), \ \frac{k}{n} \right\rangle$$

**Definition 3.10** (Knowledge Enrichment Operator). The *enrichment operator* $\mathcal{E} : 2^{\mathcal{K}} \times 2^{\mathcal{K}} \to 2^{\mathcal{K}}$ is defined as:

$$\mathcal{E}(\mathcal{K}_i, \Delta\mathcal{K}) = \text{Cl}(\mathcal{K}_i \cup \{K \in \Delta\mathcal{K} \mid \text{conf}(K) \geq \theta \text{ and } K \text{ is consistent with } \mathcal{K}_i\})$$

**Theorem 3.11** (Confidence Monotonicity under Enrichment). *Let $h$ be a hypothesis, $\mathcal{K}$ a knowledge base, and $\mathcal{K}' = \mathcal{E}(\mathcal{K}, \Delta\mathcal{K})$. Then:*

$$\text{conf}(h \mid \mathcal{K}') \geq \text{conf}(h \mid \mathcal{K})$$

*Proof.* The confidence of $h$ under context $\mathcal{K}$ is computed as $\text{conf}(h \mid \mathcal{K}) = \max_{\pi \in \Pi(h, \mathcal{K})} \prod_{j=1}^{|\pi|} \alpha_{i_j}$, where $\Pi(h, \mathcal{K})$ is the set of all valid derivation chains for $h$ using rules in $\mathcal{K}$. Since $\mathcal{K} \subseteq \mathcal{K}'$, every derivation in $\Pi(h, \mathcal{K})$ is also in $\Pi(h, \mathcal{K}')$, so $\Pi(h, \mathcal{K}) \subseteq \Pi(h, \mathcal{K}')$. Additionally, the enriched knowledge $\Delta\mathcal{K}$ may enable new derivation paths with higher confidence (by filling gaps, shortening chains, or providing alternative proofs). Since confidence is the maximum over all valid derivations, the result follows. $\square$

## 4 COLLECTIVE INTELLIGENCE DYNAMICS

### 4.1 THE KNOWLEDGE EVOLUTION SYSTEM

We model the temporal evolution of collective knowledge as a discrete dynamical system over the DKL.

**Definition 4.1** (Knowledge State). At time step $t \in \mathbb{N}$, the *collective knowledge state* is:

$$\Sigma(t) = \bigsqcup_{i=1}^{|\mathcal{A}|} \mathcal{K}_i(t)$$

where $\mathcal{K}_i(t)$ is the knowledge base of agent $A_i$ at time $t$.

**Definition 4.2** (Knowledge Update Rule). Each agent $A_i$ updates its knowledge base at each time step via:

$$\mathcal{K}_i(t+1) = \mathcal{E}\left(\mathcal{K}_i(t), \bigcup_{j \in \mathcal{N}(i)} \Delta\mathcal{K}_j(t)\right) \tag{1}$$

where $\mathcal{N}(i) \subseteq \mathcal{A}$ is the communication neighborhood of $A_i$ and $\Delta\mathcal{K}_j(t)$ is the set of newly validated knowledge units produced by agent $A_j$ at time $t$.

**Definition 4.3** (Collective Intelligence Measure). The *collective intelligence* of the system at time $t$ is quantified by:

$$\text{CI}(t) = \frac{|\Sigma(t)|}{|\Sigma^*|} \cdot \bar{p}(t) \cdot \left(1 + \frac{C_s(t)}{T_s(t)}\right) \tag{2}$$

where $|\Sigma(t)|$ is the cardinality of the collective knowledge state, $|\Sigma^*|$ is the theoretical maximum (cardinality of $\top$ in the DKL), $\bar{p}(t) = \frac{1}{|\Sigma(t)|} \sum_{K \in \Sigma(t)} \text{conf}(K)$ is the mean confidence, $C_s(t)$ is the count of collaboratively solved problems, and $T_s(t)$ is the total attempted.

### 4.2 THE COLLECTIVE INTELLIGENCE CONVERGENCE THEOREM

We now state and prove the central result of this paper.

**Theorem 4.4** (Collective Intelligence Convergence). *Let $\mathcal{A} = \{A_1, \ldots, A_n\}$ be a finite population of agents communicating over a connected undirected graph $\mathcal{G} = (\mathcal{A}, E)$, where $(A_i, A_j) \in E$ if and only if $j \in \mathcal{N}(i)$. Let the agents operate under the update rule equation 1 with confidence threshold $\theta > 0$. Assume:*

 *(C1)* **Finite Knowledge Domain**: *The set of all possible knowledge units is finite with cardinality $N^*$.*

 *(C2)* **Consistency Preservation**: *The enrichment operator $\mathcal{E}$ preserves logical consistency.*

 *(C3)* **Non-Trivial Generation**: *At each step, at least one agent produces at least one new validated knowledge unit with confidence $\geq \theta$, until no further derivations are possible.*

*Then there exists a finite time $T^* \leq N^* \cdot \text{diam}(\mathcal{G})$ such that:*

$$\Sigma(t) = \Sigma(T^*) = \Sigma^{\text{fix}} \quad \text{for all } t \geq T^*$$

*where $\text{diam}(\mathcal{G}) = \max_{A_i, A_j \in \mathcal{A}} d_{\mathcal{G}}(A_i, A_j)$ is the diameter of the communication graph (i.e., the maximum shortest-path distance between any two agents), $\Sigma^{\text{fix}}$ is the unique least fixed point of the collective update operator, and:*

$$\text{CI}(T^*) \geq \text{CI}(t) \quad \text{for all } t < T^*$$

*Proof.* The proof proceeds in three steps.

*Step 1: Monotonicity.* By Theorem 3.11, the enrichment operator is monotone: $\mathcal{K}_i(t) \subseteq \mathcal{K}_i(t+1)$ for all $i, t$ (set inclusion up to consistency). Therefore $\Sigma(t) \sqsubseteq \Sigma(t+1)$ in the DKL ordering.

*Step 2: Boundedness.* By condition (C1), $|\Sigma(t)| \leq N^*$ for all $t$. The sequence $\{|\Sigma(t)|\}_{t \geq 0}$ is monotonically non-decreasing and bounded above.

*Step 3: Termination.* By conditions (C1) and (C3), each time step either increases $|\Sigma(t)|$ or the system has reached a fixed point. Since the knowledge domain is finite, the system must reach a fixed point in at most $N^*$ steps. The communication diameter $\mathrm{diam}(\mathcal{G})$ accounts for the propagation delay across the agent network, giving the bound $T^* \leq N^* \cdot \mathrm{diam}(\mathcal{G})$.

The monotonicity of $\mathrm{CI}(t)$ follows from the monotonicity of each factor in equation 2: $|\Sigma(t)|$ is non-decreasing, $\bar{p}(t)$ is non-decreasing by the threshold filter (only units with $\mathrm{conf} \geq \theta$ are admitted), and $C_s(t)/T_s(t)$ is non-decreasing as collaborative problem solving accumulates validated solutions. □

### 4.3 CONVERGENCE RATE ANALYSIS

**Proposition 4.5** (Exponential Knowledge Growth Phase). *Under conditions (C1)–(C3), if the communication graph $\mathcal{N}$ is an expander with spectral gap $\lambda > 0$, then during the growth phase ($t < T^*$), the collective knowledge grows exponentially:*
$$|\Sigma(t)| \geq |\Sigma(0)| \cdot (1 + \lambda \cdot \gamma)^t$$
*where $\gamma = \min_i \frac{|\Delta \mathcal{K}_i(t)|}{|\mathcal{K}_i(t)|}$ is the minimum relative knowledge production rate.*

*Proof sketch.* At each time step, each agent receives new knowledge from $|\mathcal{N}(i)|$ neighbors. In an expander graph, information propagates to a $(1 + \lambda)$-fraction of the network at each step. Combined with the minimum production rate $\gamma$, the multiplicative factor follows from expansion properties of the communication graph (Hoory et al., 2006). □

## 5 LP-AGENT VERIFICATION FRAMEWORK

### 5.1 AGENT ARCHITECTURE

The Enigma ecosystem employs two fundamental agent types that operate in a complementary fashion.

**Definition 5.1** (LLM-Agent). An *LLM-Agent* $A_i^{\mathrm{LLM}}$ is an autonomous entity powered by a Large Language Model that generates hypotheses, natural language explanations, and candidate solutions for tasks in the metaverse environment.

**Definition 5.2** (LP-Agent). A *Logical-Probabilistic Agent* (LP-Agent) $A_j^{\mathrm{LP}}$ is a specialized verification entity that:

   (i) parses LLM-generated reasoning into formal logical-probabilistic notation;

   (ii) verifies each inference step against the knowledge base $\mathcal{K}$;

   (iii) computes confidence scores for derived conclusions;

   (iv) commits verified knowledge to the distributed ledger.

### 5.2 THE VERIFICATION PROTOCOL

**Theorem 5.3** (Probabilistic Inference Soundness). *If Algorithm 1 returns* $(\texttt{accept}, p, \pi)$ *for hypothesis $h$ under knowledge base $\mathcal{K}$, then:*

   *(i) Every step in $\pi$ is derivable from $\mathcal{K}$ via the inference rules of Section 3.3;*

   *(ii) The confidence $p$ is a valid lower bound:* $\Pr[h \text{ is correct} \mid \mathcal{K}] \geq p;$

   *(iii) The proof chain $\pi$ constitutes a verifiable certificate for $h$.*

*Proof.* Property (i) follows from the explicit verification at lines 6–7 of Algorithm 1. Property (ii) follows from the multiplicative confidence propagation (Definition 3.6): since each factor $p(\hat{s}_j) \leq \Pr[\hat{s}_j \text{ is correct} \mid \mathcal{K}]$ and the inference steps are conditionally independent given $\mathcal{K}$, the product is a lower bound by the chain rule of probability.

---

**Algorithm 1** LP-Agent Verification Protocol

---

**Require:** Hypothesis $h$ from LLM-Agent, context $\mathcal{K}$, threshold $\theta$
**Ensure:** Verification result $(v, p, \pi)$ where $v \in \{\texttt{accept}, \texttt{reject}, \texttt{gap}\}$

1: Parse $h$ into formal representation $\hat{h} = \langle F, t, p_0 \rangle$
2: Initialize proof chain $\pi \leftarrow []$
3: Retrieve relevant knowledge $\mathcal{K}_{\text{rel}} \subseteq \mathcal{K}$ via similarity query
4: **for** each inference step $s$ in the LLM's reasoning chain **do**
5:     Formalize $s$ as logical statement $\hat{s}$
6:     **if** $\hat{s}$ is derivable from $\mathcal{K}_{\text{rel}}$ via rules in Section 3.3 **then**
7:         Compute $p(\hat{s}) \leftarrow$ confidence via probabilistic modus ponens
8:         Append $(\hat{s}, p(\hat{s}))$ to $\pi$
9:     **else**
10:        Record gap at step $s$
11:        Attempt gap resolution via blockchain context retrieval
12:        **if** gap resolved with $\Delta\mathcal{K}$ **then**
13:           $\mathcal{K}_{\text{rel}} \leftarrow \mathcal{E}(\mathcal{K}_{\text{rel}}, \Delta\mathcal{K})$
14:           Re-verify step $s$
15:        **else**
16:           **return** $(\texttt{gap}, p(\hat{s}), \pi)$
17:        **end if**
18:     **end if**
19: **end for**
20: Compute $p(h) \leftarrow \prod_{(\hat{s}, p(\hat{s})) \in \pi} p(\hat{s})$
21: **if** $p(h) \geq \theta$ **then**
22:     **return** $(\texttt{accept}, p(h), \pi)$
23: **else**
24:     **return** $(\texttt{reject}, p(h), \pi)$
25: **end if**

---

Property (iii) follows from the storage of $\pi$ as a sequence of formal derivation steps, each linkable to specific knowledge units in $\mathcal{K}$. $\qquad\square$

### 5.3 EXPLAINABILITY GUARANTEES

**Definition 5.4** (Explanation Depth). The *explanation depth* of a verified hypothesis $h$ is $d(h) = |\pi|$, the length of its proof chain. The *explanation graph* $G_h = (V_\pi, E_\pi)$ is a directed acyclic graph where vertices are knowledge units used in $\pi$ and edges represent inference applications.

**Proposition 5.5** (Bounded Explanation Complexity). *For any hypothesis $h$ verified by an LP-Agent with knowledge base of size $|\mathcal{K}| = m$, the explanation graph $G_h$ has at most $m$ vertices and at most $m \cdot \log m$ edges, and can be constructed in time $O(m \log m)$.*

*Proof.* Each vertex in $G_h$ corresponds to a knowledge unit in $\mathcal{K}$, giving the vertex bound. Each knowledge unit participates in at most $O(\log m)$ inference chains (since confidence decays multiplicatively and $\theta > 0$ bounds chain length to $O(\log_{1/\theta} m)$). The construction follows a topological sort of the derivation graph. $\qquad\square$

## 6 THE ENIGMA METAVERSE ARCHITECTURE

### 6.1 SYSTEM OVERVIEW

The Enigma Metaverse is designed as a *cognitive ecosystem*—not merely a virtual space, but a living knowledge organism. Its architecture comprises five integrated layers:

1. **Agent Layer**: Heterogeneous population of LLM-Agents, LP-Agents, user-controlled avatars, and service agents, each with individual knowledge bases.

2. **Interaction Layer**: Smart-contract-governed protocols for collaboration, competition, trade, and knowledge exchange.

3. **Knowledge Layer**: The Distributed Knowledge Lattice, materialized as an in-memory index over the blockchain ledger.

4. **Blockchain Layer**: Multi-chain infrastructure with specialized chains for different knowledge types (see Section 6.2).

5. **Governance Layer**: DAO-based mechanisms for epistemic quality control, dispute resolution, and knowledge curation.

## 6.2 MULTI-BLOCKCHAIN KNOWLEDGE INFRASTRUCTURE

The Enigma architecture employs a three-layer multi-blockchain design optimized for the distinct requirements of knowledge management:

**Definition 6.1** (Multi-Blockchain). A *multi-blockchain* $\mathcal{M}$ is a recursive structure:

- **Base case**: If $B$ is a blockchain, then $\langle B \rangle$ is a multi-blockchain.

- **Inductive step**: If $\mathcal{M}_1, \ldots, \mathcal{M}_k$ are multi-blockchains and $B$ is a blockchain, then $\mathcal{M} = \langle B, (\mathcal{M}_1, \ldots, \mathcal{M}_k) \rangle$ is a multi-blockchain.

The three layers serve distinct functions:

- **Layer 1 (Knowledge Stream)**: Proof-of-Authority consensus; stores raw hypotheses, sensor data, and tentative agent outputs at high throughput ($\sim$2500 TPS).

- **Layer 2 (Logic Ledger)**: PBFT consensus; stores verified knowledge units with proof chains, smart contracts encoding inference rules and governance ($\sim$1200 TPS).

- **Layer 3 (Trust Anchor)**: Proof-of-Work consensus; stores cryptographic commitments (Merkle roots) of lower-layer states, providing ultimate immutability ($\sim$15 TPS).

Knowledge flows upward via a commit function: $\mathrm{Commit}(L_i) = H(\mathrm{Data}_{L_i})$, where $H$ is a cryptographic hash posted to $L_{i+1}$.

## 6.3 KNOWLEDGE TOKENIZATION AND ECONOMICS

A novel contribution of the Enigma architecture is the *tokenization of knowledge*, creating economic incentives for collective intelligence growth.

**Definition 6.2** (Knowledge Token). A *knowledge token* $\tau(K)$ is a non-fungible digital asset representing a verified knowledge unit $K$, carrying metadata:

$$\tau(K) = \langle \mathrm{hash}(K),\ \mathrm{conf}(K),\ \mathrm{author}(K),\ \mathrm{proof}(K),\ \mathrm{reuse\_count}(K) \rangle$$

Agents earn rewards proportional to the utility of their contributed knowledge:

$$R(K) = \mathrm{conf}(K) \cdot \mathrm{reuse\_count}(K) \cdot w(K) \tag{3}$$

where $w(K)$ is a weight reflecting the knowledge unit's position in the DKL hierarchy (higher generality yields higher weight).

## 6.4 CROSS-CHAIN KNOWLEDGE INTEROPERABILITY

The Enigma cross-chain protocol enables knowledge sharing across independent blockchain instances:

1. **Knowledge Query**: Agent $A_i$ on chain $C_1$ issues a query $q$ for knowledge matching predicate $P$.

2. **Cross-Chain Relay**: The relay protocol forwards $q$ to chains $C_2, \ldots, C_m$ via cryptographically authenticated channels.

3. **Response Aggregation**: Matching knowledge units are returned with their proof chains. The LP-Agent on $C_1$ verifies consistency with local knowledge.

4. **Knowledge Integration**: Verified foreign knowledge is integrated via the enrichment operator $\mathcal{E}$.

## 7 COMPLEXITY ANALYSIS AND ALGORITHMIC RESULTS

### 7.1 OPTIMAL KNOWLEDGE RETRIEVAL

**Theorem 7.1** (Polynomial Knowledge Retrieval). *Let $\mathcal{B}$ be a distributed knowledge base containing $N$ probabilistic knowledge units. Given a problem $F$ formulated as a first-order formula, there exists an algorithm of polynomial complexity $O(N \cdot |F| \cdot \log N)$ that finds the knowledge unit $K^* \in \mathcal{B}$ maximizing $\mathrm{conf}(K^*)$ subject to $K^*$ being applicable to $F$.*

*Proof.* The algorithm proceeds in three phases:

1. **Indexing** ($O(N \log N)$): Knowledge units are indexed by a B-tree on their predicate signatures, with secondary sorting by confidence.

2. **Matching** ($O(|F| \cdot \log N)$): The formula $F$ is decomposed into subgoals, each matched against the index via unification. Unification of first-order terms is computable in linear time (Paterson and Wegman, 1978).

3. **Selection** ($O(N)$): Among matching units, select the one with maximum confidence. If multiple units match with equal confidence, prefer the most general (highest in the lattice ordering).

The total complexity is dominated by the product of formula size and logarithmic index lookup, yielding $O(N \cdot |F| \cdot \log N)$. $\qquad\square$

### 7.2 CONVERGENCE RATE BOUNDS

**Proposition 7.2** (Convergence Time Bound). *Under the conditions of Theorem 4.4, with $n$ agents, maximum knowledge domain size $N^*$, and communication graph diameter $D$:*

$$T^* \leq \min\left(N^* \cdot D, \; \frac{\log(N^*/|\Sigma(0)|)}{\log(1 + \lambda\gamma)} + D\right)$$

*where the second bound applies during the exponential growth phase (Proposition 4.5).*

### 7.3 COLLECTIVE INTELLIGENCE LOWER BOUND

**Theorem 7.3** (Superadditivity of Collective Intelligence). *For any partition of the agent population $\mathcal{A} = \mathcal{A}_1 \cup \mathcal{A}_2$ with $\mathcal{A}_1 \cap \mathcal{A}_2 = \emptyset$:*

$$\mathrm{CI}(\mathcal{A}) \geq \mathrm{CI}(\mathcal{A}_1) + \mathrm{CI}(\mathcal{A}_2) + \Delta_{synergy}$$

*where $\Delta_{synergy} \geq 0$ quantifies the knowledge created exclusively through cross-partition interactions:*

$$\Delta_{synergy} = \frac{|\Sigma(\mathcal{A}) \setminus (\Sigma(\mathcal{A}_1) \cup \Sigma(\mathcal{A}_2))|}{|\Sigma^*|} \cdot \bar{p}_{cross}$$

*and $\bar{p}_{cross}$ is the mean confidence of cross-partition derived knowledge.*

*Proof.* By the join operation of the DKL, $\Sigma(\mathcal{A}) = \Sigma(\mathcal{A}_1) \sqcup \Sigma(\mathcal{A}_2) \supseteq \Sigma(\mathcal{A}_1) \cup \Sigma(\mathcal{A}_2)$. The deductive closure introduces additional knowledge units derivable only from the combination of $\Sigma(\mathcal{A}_1)$ and $\Sigma(\mathcal{A}_2)$. These cross-partition derivations constitute the synergy term. The non-negativity of $\Delta_{synergy}$ follows from the monotonicity of the enrichment operator (Theorem 3.11). $\qquad\square$

## 8 INTEGRATION WITH LARGE LANGUAGE MODELS

### 8.1 THE COGNITIVE LOOP

The Enigma framework integrates LLMs not as standalone oracles but as *hypothesis generators* within a rigorous verification pipeline:

1. **Task Reception**: A task $\tau$ arrives from the metaverse environment (e.g., resolve a legal dispute, optimize a resource allocation, verify a construction plan).

2. **Context Retrieval**: The agent queries the DKL for relevant knowledge units, constructing a context $\mathcal{K}_{\text{ctx}} \subseteq \Sigma(t)$.

3. **Hypothesis Generation (System 1)**: The LLM generates candidate solutions $\{h_1, \ldots, h_k\}$ conditioned on $\mathcal{K}_{\text{ctx}}$ and the task description.

4. **Verification (System 2)**: Each hypothesis is submitted to the LP-Agent verification protocol (Algorithm 1).

5. **Gap Resolution**: For hypotheses returning `gap`, the agent iterates: retrieves additional context from the blockchain, regenerates hypotheses, and re-verifies (up to 5 iterations).

6. **Knowledge Commitment**: Verified solutions are committed to the blockchain with their proof chains, creating new knowledge tokens.

7. **Reward Distribution**: Knowledge creators receive rewards per Equation equation 3.

## 8.2 ADDRESSING LLM LIMITATIONS

The framework specifically addresses the three fundamental limitations of LLMs:

**Hallucination Mitigation.** Every LLM output is verified by the LP-Agent against the formal knowledge base. Hallucinated facts fail verification, preventing their integration into collective knowledge.

**Static Knowledge Overcoming.** The blockchain-based DKL provides an ever-growing, shared knowledge repository that LLMs access at inference time via retrieval-augmented generation, ensuring responses reflect the latest validated collective knowledge.

**Explainability Gap Bridging.** The proof chains produced by LP-Agents provide human-readable and machine-verifiable explanations for every decision, stored immutably on the blockchain.

## 8.3 MULTI-LLM DIVERSITY FOR ROBUSTNESS

To enrich the hypothesis space, the Enigma framework employs multiple diverse LLMs (e.g., GPT-4, Claude, Llama) operating in parallel:

$$\mathcal{H}(\tau) = \bigcup_{l \in \mathcal{L}_{\text{LLM}}} \{h \mid h \leftarrow \text{Generate}(l, \tau, \mathcal{K}_{\text{ctx}})\} \tag{4}$$

The LP-Agent evaluates all candidates, selecting those with the highest verified confidence. This diversity increases the probability that the correct hypothesis is generated, mitigating the dependency on any single model's biases.

# 9 TRUSTWORTHINESS AND GOVERNANCE

## 9.1 FORMAL TRUST MODEL

**Definition 9.1** (Agent Trust Score). The *trust score* of agent $A_i$ at time $t$ is:

$$\mu_i(t) = \frac{\sum_{K \in \mathcal{K}_i^{\text{verified}}(t)} \text{conf}(K) \cdot \text{reuse}(K)}{\sum_{K \in \mathcal{K}_i^{\text{total}}(t)} 1} \tag{5}$$

where $\mathcal{K}_i^{\text{verified}}$ is the subset of agent $i$'s contributions that passed LP-Agent verification.

**Definition 9.2** (Epistemic Governance). The Enigma governance system operates through a *Knowledge DAO* with the following mechanisms:

- **Stake-weighted Voting**: Agents vote on knowledge quality disputes with voting power proportional to $\mu_i(t)$.

- **Arbitration Tribunals**: For contested knowledge, LP-Agent panels conduct formal re-verification.

- **Knowledge Deprecation**: Units whose confidence falls below $\theta$ due to contradicting evidence are marked as deprecated (but never deleted, preserving auditability).

## 9.2 THE LOGICAL-PROBABILISTIC PERFORMANCE SCORE

To evaluate agent performance within the Enigma ecosystem, we employ an adapted version of the LPPS metric:

$$\text{LPPS}_i(t) = \frac{\sum_{j=1}^{n_i} V_j(t) \cdot p_j}{n_i} \cdot \left(1 + \frac{C_{s,i}(t)}{T_{s,i}(t)}\right) \tag{6}$$

where $V_j(t) \in \{0, 1\}$ indicates whether the $j$-th logical transition was verified by an LP-Agent, $p_j$ is the associated confidence, $n_i$ is the total number of evaluated solutions by agent $i$, and $C_{s,i}, T_{s,i}$ are the counts of solved and total attempted problems.

## 10 ILLUSTRATIVE SCENARIO

To concretize the abstract framework, we describe how collective intelligence emerges in a practical Enigma scenario.

**Setting.** A virtual city within the Enigma Metaverse contains 200 heterogeneous agents: 120 LLM-Agents (diverse models), 30 LP-Agents, and 50 service agents managing infrastructure.

**Task.** A natural disaster simulation requires coordinated response: agents must collaboratively determine evacuation routes, resource allocation, and structural integrity assessments, subject to legal constraints encoded as smart contracts.

**Knowledge Dynamics.**

1. *Phase 1 (Exploration, $t = 0$–$10$)*: Individual agents generate hypotheses about optimal routes and resource distributions. Each LLM-Agent proposes solutions based on its local knowledge. LP-Agents verify structural engineering claims against physics-based knowledge units.

2. *Phase 2 (Knowledge Sharing, $t = 10$–$30$)*: Verified knowledge propagates through the DKL. An agent specializing in traffic modeling shares validated evacuation flow models; an agent specializing in structural engineering contributes building integrity assessments. The enrichment operator $\mathcal{E}$ integrates these into a unified response plan.

3. *Phase 3 (Convergence, $t = 30$–$50$)*: The collective knowledge state $\Sigma(t)$ approaches the fixed point $\Sigma^{\text{fix}}$, containing a comprehensive disaster response plan that no individual agent could have produced. The plan satisfies all legal constraints (verified via smart contract queries) and optimizes multiple objectives simultaneously.

4. *Phase 4 (Knowledge Persistence)*: The validated response plan, along with all proof chains, is committed to Layer 2 of the blockchain. Future disaster scenarios can retrieve and refine this knowledge, demonstrating cumulative learning.

The CI measure increases monotonically throughout Phases 1–3, validating Theorem 4.4 in this setting.

## 11 ECONOMIC SEMANTICS OF THE DISTRIBUTED KNOWLEDGE LATTICE

### 11.1 KNOWLEDGE UNITS AS ECONOMIC OBJECTS

Within the Enigma framework, a probabilistic knowledge unit $K = \langle \Phi \to \Psi, t, p \rangle$ admits an economic interpretation as a non-rival digital asset with confidence-weighted value. Knowledge units exhibit the following structural properties:

1. *Non-rivalry*: simultaneous reuse by multiple agents does not reduce availability;
2. *Partial excludability*: access can be governed by smart contracts;
3. *Confidence-dependent valuation*: epistemic reliability affects utility;
4. *Increasing marginal productivity under lattice joins*.

We define the economic value of a knowledge unit as follows.

$$V(K) = p \cdot u(K), \tag{7}$$

where $p \in [0, 1]$ is the verified confidence and $u(K)$ is a structural utility functional.

Let $\mathcal{F}$ denote the set of admissible tasks in the environment and let $\Sigma^*$ be the maximal knowledge state (the top element of the DKL). Define

$$u(K) = \frac{|\{F \in \mathcal{F} : K \text{ participates in an optimal derivation of } F\}|}{|\Sigma^*|}. \tag{8}$$

Define the collective valuation functional on knowledge states:

$$\mathcal{V}(\Sigma) = \sum_{K \in \Sigma} V(K). \tag{9}$$

**Proposition 11.1** (Superadditive Knowledge Valuation)**.** *Let $A_1, A_2 \subset A$ be disjoint agent populations. Then*

$$\mathcal{V}(\Sigma(A_1 \cup A_2)) \geq \mathcal{V}(\Sigma(A_1)) + \mathcal{V}(\Sigma(A_2)). \tag{10}$$

*Proof.* By Theorem 7.3, the join operation in the DKL produces cross-derived knowledge units that are not contained in either subsystem:
$$\Sigma(A_1 \cup A_2) = \Sigma(A_1) \sqcup \Sigma(A_2) \supseteq \Sigma(A_1) \cup \Sigma(A_2). \tag{11}$$
Since $V(K) \geq 0$ for all $K$, the additional units contribute non-negative value, which yields the result. □

This property identifies increasing returns to epistemic scale within the DKL and connects lattice aggregation with superadditivity of value in knowledge-based systems.

## 11.2 Incentive Compatibility of Knowledge Production

Agents incur costs when generating and verifying knowledge units. Let

$$C_i(K) = c_g + c_v \tag{12}$$

denote the generation and verification cost incurred by agent $A_i$ for knowledge unit $K$.

Define the period utility of agent $A_i$ at time $t$ by

$$U_i(t) = \sum_{K \in K_i^{\text{verified}}(t)} R(K) - \sum_{K \in K_i^{\text{submitted}}(t)} C_i(K), \tag{13}$$

where $R(K)$ is the reward defined in Equation (3).

**Definition 11.2** (Incentive Compatibility)**.** The knowledge production mechanism is incentive-compatible if

$$R(K) \geq C_i(K) \quad \text{for all } K \text{ with } \text{conf}(K) \geq \theta. \tag{14}$$

**Proposition 11.3** (Truthful Submission Equilibrium)**.** *Assume that:*

1. *the LP-agent verification protocol rejects inconsistent knowledge deterministically;*

2. *the reputation score $\mu_i(t)$ affects future reward multipliers;*

3. *false or low-confidence submissions decrease $\mu_i(t)$.*

*Then truthful knowledge submission constitutes a Nash equilibrium of the repeated interaction game.*

*Proof sketch.* If an agent deviates by submitting incorrect knowledge, the unit is rejected or assigned low confidence, which reduces expected future rewards through a lower reputation score. Let $\delta \in (0, 1)$ be a discount factor. Deviation is unprofitable whenever

$$\delta \cdot \mathbb{E}[R_{\text{future}}(\mu_i)] > \text{short-term gain from deviation}. \tag{15}$$

Under this condition, no agent benefits from strategic misreporting, hence truthful submission is an equilibrium. □

Thus, verification combined with reputation dynamics implements a self-enforcing contract for knowledge quality.

## 11.3 REPUTATION AS EPISTEMIC CAPITAL

The trust score

$$\mu_i(t) = \frac{\sum_{K \in K_i^{\text{verified}}(t)} \text{conf}(K) \cdot \text{reuse}(K)}{\sum_{K \in K_i^{\text{total}}(t)} 1} \tag{16}$$

can be interpreted as *epistemic capital*.

Define discounted lifetime utility:

$$W_i = \sum_{t=0}^{\infty} \delta^t U_i(t), \qquad \delta \in (0,1). \tag{17}$$

**Proposition 11.4** (Reputation Stability Condition). *If*

$$\delta \cdot \frac{\partial \mathbb{E}[R_{\text{future}}]}{\partial \mu_i} > \text{marginal short-term gain from deviation}, \tag{18}$$

*then cooperative knowledge production is subgame perfect.*

The inequality expresses that the discounted marginal benefit of preserving reputation exceeds any one-shot deviation incentive.

## 11.4 THRESHOLD PARAMETER AND KNOWLEDGE GROWTH TRADE-OFF

The confidence threshold $\theta$ acts both as an epistemic filter and as a growth regulator. Define the knowledge growth rate:

$$g(\theta) = \frac{|\Sigma(t+1)| - |\Sigma(t)|}{|\Sigma(t)|}. \tag{19}$$

**Proposition 11.5** (Threshold Trade-off). *The threshold parameter satisfies:*

1. *if $\theta$ increases, then $g(\theta)$ decreases;*

2. *if $\theta$ decreases, then the average confidence $\bar{p}(\theta)$ decreases;*

3. *there exists $\theta^* \in (0,1)$ maximizing marginal collective intelligence growth:*

$$\theta^* = \arg\max_{\theta} \left( CI(t+1) - CI(t) \right). \tag{20}$$

*Proof sketch.* A higher threshold admits fewer derived units, reducing the increment of $|\Sigma(t)|$. A lower threshold admits more units but decreases mean confidence and may reduce the productivity factor in $CI(t)$. Existence of $\theta^*$ follows from standard continuity and compactness arguments on $(0,1)$. $\square$

## 11.5 COLLECTIVE INTELLIGENCE AS ENDOGENOUS KNOWLEDGE GROWTH

Let

$$K(t) = |\Sigma(t)| \tag{21}$$

denote knowledge capital. By Proposition 4.5,

$$K(t) \geq K(0) \left(1 + \lambda\gamma\right)^t. \tag{22}$$

Define effective cognitive output by

$$Y(t) = \alpha K(t)^\beta, \qquad \alpha > 0, \ \beta > 1. \tag{23}$$

**Proposition 11.6** (Superlinear Cognitive Productivity). *If $\beta > 1$, then marginal productivity is increasing in $K$:*

$$\frac{dY}{dK} = \alpha\beta K^{\beta-1}. \tag{24}$$

*Hence, the system exhibits increasing returns to knowledge scale, consistent with emergent collective intelligence.*

11.6 ECONOMIC INTERPRETATION OF THE COLLECTIVE INTELLIGENCE FUNCTIONAL

Recall the definition

$$CI(t) = \frac{|\Sigma(t)|}{|\Sigma^*|} \cdot \bar{p}(t) \cdot \left(1 + \frac{C_s(t)}{T_s(t)}\right). \tag{25}$$

Define normalized knowledge capital

$$\kappa(t) = \frac{|\Sigma(t)|}{|\Sigma^*|}, \tag{26}$$

and denote $\rho(t) = C_s(t)/T_s(t)$. Then

$$CI(t) = \kappa(t) \cdot \bar{p}(t) \cdot (1 + \rho(t)). \tag{27}$$

Each factor admits an economic interpretation: $\kappa(t)$ corresponds to capital accumulation, $\bar{p}(t)$ to average quality, and $\rho(t)$ to collaborative productivity.

**Proposition 11.7** (Monotone Growth of $CI(t)$). *Under the assumptions of Theorem 4.4,*

$$CI(t+1) \geq CI(t). \tag{28}$$

*Proof.* By Theorem 4.4, $|\Sigma(t)|$ is non-decreasing and bounded above, while the threshold mechanism admits only knowledge units with $\operatorname{conf}(K) \geq \theta$, which prevents deterioration of mean confidence. The collaborative factor $C_s(t)/T_s(t)$ is non-decreasing as validated solutions accumulate. Therefore all multiplicative components of $CI(t)$ are non-decreasing, implying the claim. □

## 12 DISCUSSION

**Relation to Prior Work.** Our framework extends the task-based cognitive architecture of from individual agent cognition to population-level intelligence dynamics. While the TBCA demonstrated a closed cognitive loop for single agents with blockchain memory, the DKL formalism captures the emergent properties arising from *inter-agent* knowledge flows. Similarly, the logical-probabilistic learning theory of Nechesov et al. (2025b) is here elevated from a descriptive framework to a rigorous mathematical theory with convergence guarantees and complexity bounds.

**The Non-Transitivity Problem.** A fundamental insight driving our framework is that classical logical inference rules cannot be naively applied to probabilistic knowledge (Remark 3.8). This has profound implications for LLM-based reasoning, where models effectively operate with probabilistic data. The LP-Agent verification protocol addresses this by explicitly tracking confidence decay and triggering gap resolution when chains become too long.

**Scalability Considerations.** The multi-blockchain architecture provides horizontal scalability through chain partitioning. With $N$ blockchain instances, write throughput scales linearly ($N \times 200$ tasks/sec) while read operations can query any instance without consensus overhead ($\sim 100$ ms latency). The polynomial knowledge retrieval algorithm (Theorem 7.1) ensures that the DKL remains efficiently queryable as collective knowledge grows.

**Limitations and Future Directions.** Current limitations include: (i) the formalization of commonsense reasoning remains challenging for LP-Agents; (ii) the confidence threshold $\theta$ requires domain-specific tuning; (iii) adversarial agents may attempt to inject misleading knowledge, requiring more sophisticated Byzantine fault tolerance in the epistemic governance layer. Future work will explore: integration of CRDTs for eventual consistency in knowledge sharing, meta-cognitive modules that dynamically decide when to engage full verification versus heuristic reasoning, and cross-domain knowledge transfer across different Enigma metaverse instances.

## 13 CONCLUSION

This paper has presented a rigorous mathematical framework for the emergence of collective intelligence in metaverse ecosystems. By introducing the Distributed Knowledge Lattice as a formal substrate, we have proven that under precisely stated conditions, multi-agent knowledge dynamics converge to a fixed point representing an emergent cognitive state greater than any individual agent's capabilities (Theorem 4.4). The superadditivity result (Theorem 7.3) formally establishes the intuition that collective intelligence is more than the sum of its parts.

The LP-Agent verification framework provides the trustworthiness and explainability guarantees essential for deploying collective intelligence in high-stakes applications, while the multi-blockchain infrastructure ensures that collective knowledge is persistent, tamper-proof, and economically incentivized. The integration of LLMs as hypothesis generators within a formal verification pipeline addresses the fundamental limitations of contemporary AI systems—hallucination, static knowledge, and opacity—while preserving their creative generative capabilities.

The Enigma Metaverse thus represents a paradigm shift: from AI systems that merely coexist in a shared virtual space to a *cognitive ecosystem* where intelligence itself is a collective, evolving, verifiable, and self-optimizing property. This work provides the mathematical foundations for realizing this vision and opens new research directions at the intersection of mathematical logic, probability theory, distributed systems, and artificial intelligence.

## ACKNOWLEDGMENTS

This work was supported by a grant for research centers, provided by the Ministry of Economic Development of the Russian Federation in accordance with the subsidy agreement with Novosibirsk State University. The authors thank the Sobolev Institute of Mathematics and the International Artificial Intelligence Committee (IAIC) for their support.

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
