# OpenReview forum: "The Evolution of Mind: Emergence of Collective Intelligence  through Logical-Probabilistic Knowledge Dynamics  in Multi-Agent Metaverse Ecosystems"
_mathai.club/MathAI/2026/Conference — 2026 Oral_

### Official Review · Reviewer_HUKK · 2026-03-12
**Review for the paper "The Evolution of Mind: Emergence of Collective Intelligence through Logical-Probabilistic Knowledge Dynamics in Multi-Agent Metaverse Ecosystems"**

**Rating:** 6
**Confidence:** 4

**Review:**

Pros:

1. This paper introduces a new and new AI Enigma framework—a formal theory of collective intelligence emergence in metaverse ecosystems, grounded in a novel logical-probabilistic learning theory.

2. This work is in the typical style of the staff of the mathematical institute. The authors prove various theorems and formulate assumptions.

3. Of course,the work is devoted to development in the field of AI, so I think the article can be accepted.

Cons:
1. It is hard to determine the practical value of these math theorems.
It seemed that the theorems are similar to the old mean theorem—we know that this mean exists, but it is practically easier to integrate a function.

2. The first paragraph of the conclusions, in my opinion,is a simple platitude.
We already know that if you unscrew the wheel of a car,it will not go.

3. The second and third paragraphs claim a lot.
It is quite possible that the theorems are correct, but their field of application is outside of real practice.

4. It is advisable to present the architecture of THE ENIGMA METAVERSE  in the form of a block diagram.

5. The authors claim that the Enigma framework employs multiple diverse LLMs (e.g., GPT-4, Claude, Llama).
But it is impossible to verify this.

6. The authors formulate a problem statement with a virtual city and 200  heterogeneous agents, but they do not present the results (value of metrics, the Graphs for reward function behavior solution, analysis of the influence of model parameters on the prediction result).

7. The article says nothing about the software implementation of this solution.
It is not clear from the article whether this framework exists in practice.

8. The authors significantly violated the required number of pages.

---

### Official Review · Reviewer_LGcy · 2026-03-12
**Strong accept**

**Rating:** 8
**Confidence:** 4

**Review:**

The manuscript proposes a perspective formal theory of collective intelligence emergence in metaverse ecosystems, grounded in a novel logical-probabilistic learning theory. In this theory, named the Enigma, the author defines a Distributed Knowledge Lattice over multi-agent interactions. He proves that, under precisely stated monotonicity and convergence conditions, the collective knowledge of an agent population forms a complete lattice whose least upper bound represents an emergent cognitive state unreachable by any individual agent. The dynamics of knowledge creation, verification, and propagation through a system of Logical-Probabilistic Agents is considered with appropriate mathematical tools. Interesting results are presented at the final stage, for example, the following: Probabilistic Inference Soundness Theorem guaranteeing that confidence propagation through distributed reasoning chains preserves logical consistency; a polynomial-time algorithm for optimal knowledge retrieval from the distributed lattice. The manuscript stares that the proposed synthesis of mathematical logic, probability theory, LLM-based hypothesis generation, and blockchain-secured knowledge persistence provides a rigorous foundation for building self-optimizing, trustworthy, and explainable collective intelligence in virtual worlds.
1.	Mathematical Rigor: high.
2.	Novelty & Contribution: good.
3.	Relevance to MathAI: very high.
4.	Technical Quality: good.
5.	Clarity & Presentation: good.
6.	AI-Generation Risk: very low.

---

### Decision · Program_Chairs · 2026-03-14

**Decision:**

Accept (Oral)

**Comment:**

Dear Author(s),

On behalf of the Program Committee of the International Conference on Mathematics of Artificial Intelligence (MathAI 2026), we are pleased to inform you that your paper has been accepted for an oral presentation at MathAI 2026.

Your paper was evaluated through a rigorous two-stage review process involving both automated screening and expert review by members of the Program Committee. The reviewers recognized the quality and contribution of your work.

Presentation details:

- Format: Oral presentation (15–20 minutes + 5 minutes Q&A)
- Mode: You may present either in person (offline) at the conference venue in Sirius, Russia, or remotely via Zoom. Please indicate your preferred mode when confirming your participation.
- Conference dates: Marh 30 - April 3, 2026
- Website: https://mathai.club

Next steps:

1. Please confirm your participation and presentation mode by replying to this email mathai.club@yandex.ru no later than March 15, 2026 18:00 Moscow time.
2. If you plan to attend in person, the organizing committee will provide accommodation details separately.
3. Please prepare your final camera-ready manuscript according to the formatting guidelines available at https://mathai.club and upload it to OpenReview by March 15, 2026 18:00 Moscow time.

Should you have any questions regarding the program, logistics, or your presentation slot, please do not hesitate to contact us.

We look forward to your contribution to MathAI 2026.

With kind regards,

MathAI 2026 Program Committee
International Conference on Mathematics of Artificial Intelligence
https://mathai.club
OpenReview: https://openreview.net/group?id=mathai.club/MathAI/2026/Conference
Telegram: https://t.me/MathAI_club
Email: mathai.club@yandex.ru